# Sen2Grass: A Cloud-Based Solution to Generate Field-Specific Grassland Information Derived from Sentinel-2 Imagery

**Tom Hardy** [1,*] **, Lammert Kooistra** [1] **, Marston Domingues Franceschini** [1] **, Sebastiaan Richter** [2] **, Erwin Vonk** [3] **, Gé van den Eertwegh** [4] **and Dion van Deijl** [4]

1 Laboratory of Geo-Information Science and Remote Sensing, Wageningen University, P.O. Box 47, 6700 AA Wageningen, The Netherlands; lammert.kooistra@wur.nl (L.K.); marston.franceschini@wur.nl (M.D.F.)
2 Versuchs-und Bildungszentrum Landwirtschaft Haus Riswick, Elsenpass 5, 47533 Kleve, Germany; sebastiaan.richter@gmx.de
3 StellaSpark, Furkabaan 60, 3524 ZK Utrecht, The Netherlands; erwin.vonk@stellaspark.com
4 KnowH2O, Watertorenweg 12, 6571 CB Berg en Dal, The Netherlands; eertwegh@knowh2o.nl (G.v.d.E.); deijl@knowh2o.nl (D.v.D.)
* Correspondence: tom.hardy@wur.nl

**Abstract:** Grasslands are important for their ecological values and for agricultural activities such as livestock production worldwide. Efficient grassland management is vital to these values and activities, and remote sensing technologies are increasingly being used to characterize the spatiotemporal variation of grasslands to support those management practices. For this study, Sentinel-2 satellite imagery was used as an input to develop an open-source and automated monitoring system (Sen2Grass) to gain field-specific grassland information on the national and regional level for any given time range as of January 2016. This system was implemented in a cloud-computing platform (StellaSpark Nexus) designed to process large geospatial data streams from a variety of sources and was tested for a number of parcels from the Haus Riswick experimental farm in Germany. Despite outliers due to fluctuating weather conditions, vegetation index time series suggested four distinct growing cycles per growing season. Established relationships between vegetation indices and grassland yield showed poor to moderate positive trends, implying that vegetation indices could be a potential predictor for grassland biomass and chlorophyll content. However, the inclusion of larger and additional datasets such as Sentinel-1 imagery could be beneficial to developing more robust prediction models and for automatic detection of mowing events for grasslands.

**Keywords:** sen2grass; sentinel-2; stellaspark; nexus; grassland monitoring; time series; vegetation indices; cloud cover

## 1. Introduction

About one-third of the earth's terrestrial surface is covered by grassland ecosystems, making them one of the most commonly occurring land use types worldwide [1]. In addition to preserving biodiversity and supporting ecological processes [2,3], grasslands are vital for global food security, since they provide an important source for agricultural activities such as livestock production [4,5]. These practices are expected to keep increasing in the future as a result of an expanding population and growing welfare, leading to an increased competition for resources such as land, water and energy [6]. Together with global climate change, these processes have led to degradation of grasslands in both arid and semi-arid regions and are therefore being considered a global threat to grassland biodiversity and food production [7,8]. For instance, since the 1930s, most of Europe's semi-natural grassland areas have been lost or at least modified by human activity, by means of intensification or abandonment of land [8–10]. In addition, by 2005 about 43 million hectares of the Eurasian steppe and eighty percent of North America's grasslands had already been

transformed into cropland, while 60 to 80 percent of South America's grasslands had been degraded completely [11].

In the past decades, various governmental programs have been initiated for the regulation of rural development, such as the Common Agricultural Policy (CAP) aimed at increasing crop production in a sustainable way and providing farmers with financial support [8], and the designation of Natura 2000 areas, a European network of protected nature areas in order to preserve biodiversity [12]. Since permanent grasslands cover a large proportion of Europe's land surface, the CAP invests a lot in creating awareness on effective grassland management, which is important to optimize use intensity of grasslands and to cultivate those lands in a sustainable and cost-effective way, while maintaining and improving its ecological values [12,13]. To support these management practices, as well as other activities such as identifying grassland species diversity and insect carrying capacity, it is of vital importance to monitor grasslands in an effective way [4,14]. Moreover, characterization the spatiotemporal variation of grasslands is beneficial to understanding soil and vegetation properties such as nitrogen balance and crop health [15]. In the past, these characterizations have frequently been based on farmer knowledge and descriptive statistics obtained by means of field campaigns. Although these data could be very informative, they are often laborious and time-consuming to collect, and are therefore increasingly being complemented by innovative methods such as remote and proximal sensing, which are based on data from satellite and airborne based platforms and field-specific sensors [16–18].

In recent years, various studies have shown that multispectral and multitemporal remote sensing are promising ways of grassland monitoring as a support for grassland management. For instance, two studies investigated ecological indicators of grassland based on multitemporal RapidEye satellite images and succeeded in distinguishing mowing regimes of grasslands in Switzerland and high nature value grasslands from landscapes in Germany between pairs of images, respectively [19,20]. For another Swiss project, habitat quality prediction models were developed based on different *NDVI* metrics derived from Landsat imagery, and it was found that abandoned grasslands could be well predicted, while the presence of pastures and meadows was harder to characterize [21]. It was therefore proposed to use a combination of vegetation index metrics, and to include imagery from satellite platforms with high spatial and temporal resolutions such as ESA's Sentinel-2 sensing platform.

The Sentinel-2 constellation is a pair of satellites (S2A and S2B, launched in 2015 and 2017, respectively), sensing the earth in parallel with multispectral sensors as part of ESA's Copernicus earth observation program aimed at monitoring climate change and improving environmental management worldwide [22,23]. Its data have been available online by an open source data platform since January 2016 [24]. Several studies have included Sentinel-2 imagery in their analyses, for instance to assess its suitability to distinguish mowing regimes of grasslands in Central Europe, and their results suggested that Sentinel-2 time series could be very useful to differentiate between land use intensities and mowing frequencies on the national and international levels, given that appropriate processing methods such as an accurate cloud-masking algorithm would be used [25,26]. Another study confirmed Sentinel-2's expediency from an ecological point of view by applying both Sentinel-1 and Sentinel-2 data in combination with field measurements to predict grassland biodiversity and finding high prediction accuracies for a number of species-diversity indices based on those prediction models [14].

More and more remote sensing data as described in these papers are being collected each year, and in order to provide farmers and other stakeholders with useful information to support their grassland management practices, there is a need for safe, automated and large-scale remote sensing data-processing platforms [27]. One such platform is the AgroDataCube developed by Wageningen Environmental Research, aimed at making large collections of remote sensing, governmental, meteorological and other big datasets available for various applications such as vegetation monitoring and crop modeling [28].

Other commercially based or open-source platforms with the intention to provide, process and analyze remote sensing and environmental data are Amazon Web Services (AWS), Microsoft Azure and Google Earth Engine (GEE), which incorporate open datasets from satellite platforms such as Landsat, Sentinel, NOAA and MODIS [29–31]. The current paper was written within the framework of the SPECTORS project aimed at developing applications for precision farming, nature conservation and environmental protection with help of drone- and sensing technology [32]. For that project, the decision was made to develop alternative applications independent from platforms such as AWS, Azure or GEE, in order to have more control on data storage and processing algorithms as part of the project's value chain. Therefore, this paper describes the development of an open-source and automated grassland monitoring system to provide farmers, consultants, researchers and other stakeholders with up-to-date information on the condition and growth of their grasslands across a growing season, based on data from Sentinel-2 imagery.

An earlier paper on this topic focused on developing and testing such algorithm on the farm level by determining vegetation indices and time series for a limited number of grassland parcels in Kleve, Germany as a case study [33]. The main goal of the current paper is to present a framework to make this algorithm applicable to gain field-specific grassland information on the national and regional level at any given time range from January 2016 onwards; this was the moment when Sentinel-2 data went online. One way to achieve this goal is by upscaling this system to an online cloud-computing environment with the capacity to handle large data streams in a fast and automated way. The alias chosen for this grassland monitoring system is 'Sen2Grass', which stands for Sentinel-2 data to Grassland information. Two additional goals are to compare field-specific vegetation indices to meteorological data to assess whether variations in grassland development could be explained by temperature and precipitation patterns, and to establish relationships between vegetation indices and crop yield data, in order to evaluate whether vegetation indices could be a good predictor for grassland biomass and/or chlorophyll levels across time.

The development of the Sen2Grass algorithm is covered in the next section, and the results are described in Section 3: first a general overview is given on what kinds of output are included in the Nexus Graphical User Interface, and secondly, the results of Haus Riswick's case study to test the Sen2Grass algorithm are presented. The limitations and interpretation of the results are covered in the discussion, and the paper ends with the most important conclusions that were drawn from this research.

## 2. Materials and Methods

The workflow of the Sen2Grass algorithm described in this paper was written in Python, and the building blocks of code were mostly written as Python functions that were stored in separate modules. The processing chain contains various processes for harvesting, processing, calculating and uploading data from and to a central database. A detailed description of these steps is offered in Section 2.2, and the Python script to develop this processing chain is published on the first author's GitHub page, in the Spectors Project repository [34].

### 2.1. Geographical Focus

At the moment of implementing the first version of Sen2Grass in Nexus, the geographical focus was on 28 Sentinel-2 tiles covering The Netherlands, Belgium and a part of France and Germany (Figure 1). The extents of this study area are in WGS84 coordinates ranging from 50°8′48.3″ N 2°12′25.3″ E (near Amiens, France) to 53°40′24.3″ N 9°47′59.7″ E (close to Hamburg, Germany), causing some tiles on the edges of this study area to be trimmed. However, the Nexus platform is not limited to this geographical area; StellaSpark provides solutions to zoom in on any geographical area on the planet. Hence, Sentinel-2 data from other parts of the world are accessible to be used as input in an algorithm such as Sen2Grass as well.

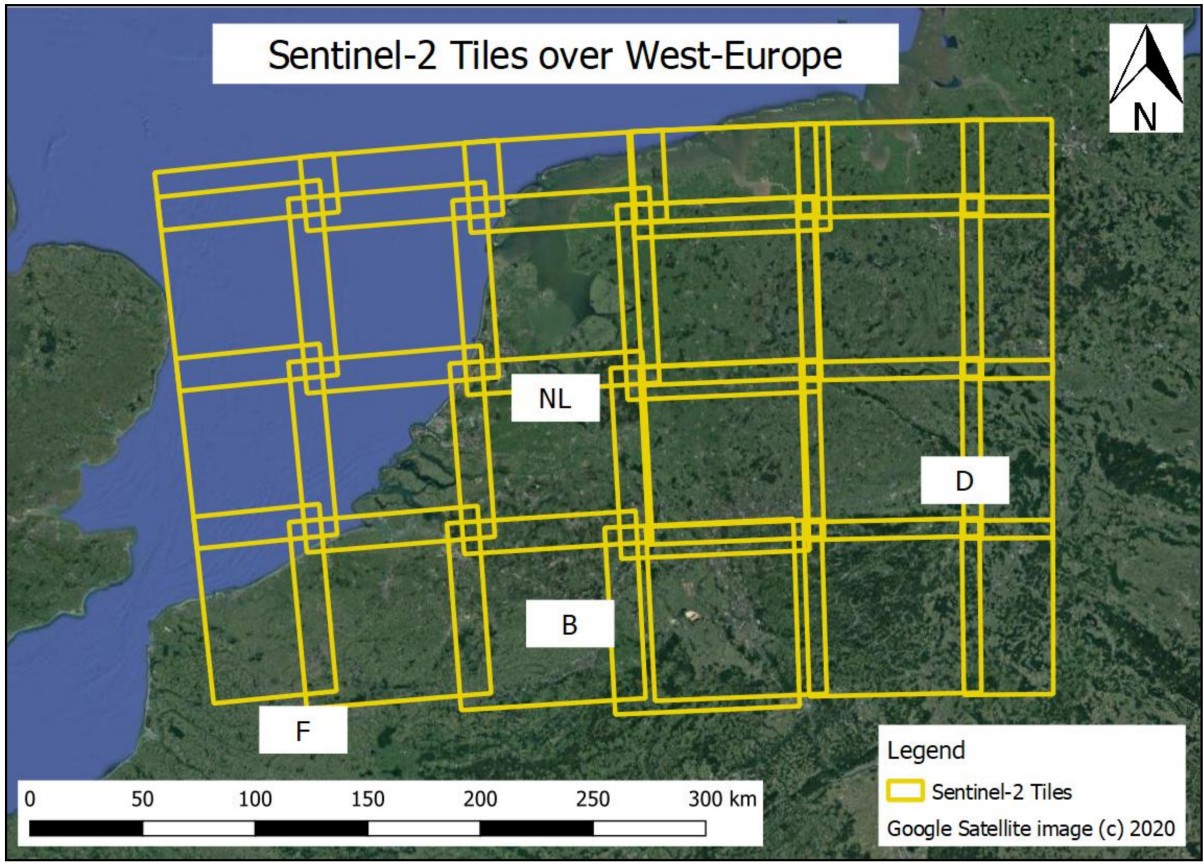

**Figure 1.** Extents of study area including all 28 Sentinel-2 tiles covering The Netherlands (NL), Belgium (B) and a part of France (F) and Germany (D).

*2.2. Development of the Sen2Grass Processing Chain*

Sen2Grass' processing chain is outlined in Figure 2, and the algorithm has been implemented in Nexus, a database platform that has been developed by the company StellaSpark, and combines data from different external sources into one storage, processing and visualization environment [35]. Examples of such data sources are landcover data retrieved from satellites, public infrastructure, cadastral parcels and weather data from various meteorological institutes worldwide. Assembling these kinds of data into one storage and processing environment allows for more efficient data management, visualization and maintenance, making such platforms user-friendly for stakeholders such as utility services, governmental agencies and academic institutions for data-driven decision making [35]. The Nexus platform makes uses of PostgreSQL with a PostGIS extension in order to store the most recent versions of harvested data from different data sources in the database. PostgreSQL is an open-source relational database server, while PostGIS is an open source spatial database extension able to contain various kinds of spatial data [36]. The structure of the database consisted of several schemas: tree-based structures hosting relational tables containing various kinds of metadata about real-world spatiotemporal datasets, ready to be harvested or appended by means of different kinds of SQL statements.

In order to develop the Sen2Grass algorithm, two main data sources linked to the Nexus platform were accessed to provide input data. The first data source was ESA's Copernicus Open Access Hub [24] providing the user with Sentinel-2 images, alternatively called Sentinel tiles. Each tile was stored in raster format, had an area of $100 \times 100$ km$^2$, and was acquired in spatial resolutions of 10 m, 20 m and 60 m, with 13 spectral bands in total ranging from RGB (visual color bands) to short wave infrared (SWIR) [37]. The second data source was the Basisregistratie Gewaspercelen (BRP), a database that contains all attributes including polygon geometry of all agricultural parcels across the Netherlands [38]. The

BRP is provided by Dutch geodata-portal Publieke dienstverlening op de kaart (PDOK), and is updated on a yearly basis [39]. Both the Sentinel-2 data and the parcel data sources were supplemented with metadata collected in relational tables necessary to retrieve the actual and corresponding datasets from the Nexus database. Among other data, the raster table contains metadata about all Sentinel-2 bands for each tile, while the plant_cover table includes metadata about parcels from a variety of arable crops. Their most important field names and data types are listed in Table 1.

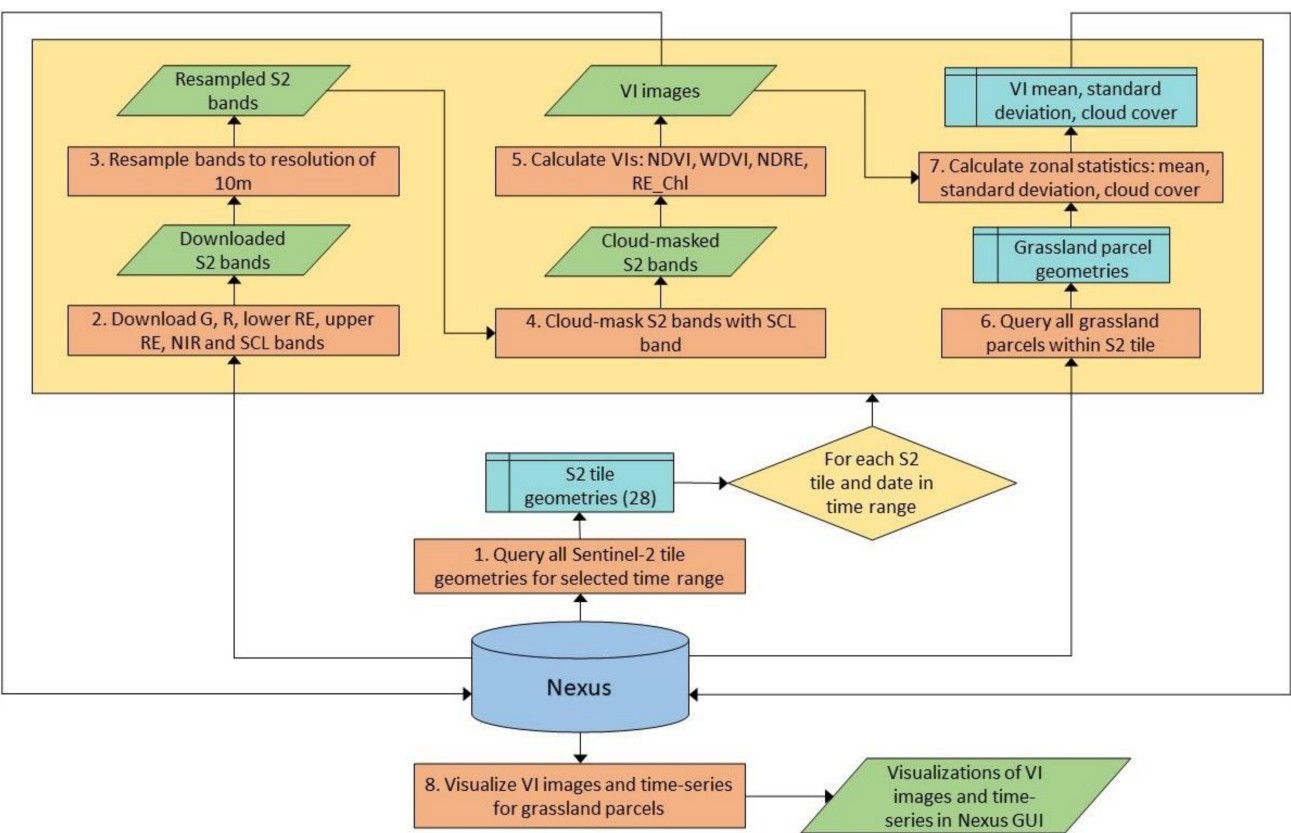

**Figure 2.** Flowchart containing processes for querying, harvesting, pre-processing, uploading, and visualizing Sentinel-2 (S2) data and vegetation indices (VI) for each S2 tile of $100 \times 100 \text{ km}^2$ and date in time range, as part of the Sen2Grass system, with the Nexus server as a central geodatabase.

**Table 1.** Most important field names including data types of the raster and plant_cover tables (respectively containing metadata about Sentinel-2 and parcel data) stored in the Nexus database, used as information in SQL statements to be able to access the actual and corresponding datasets.

| Raster Table | | Plant_Cover Table | |
|---|---|---|---|
| attribute | data type | attribute | data type |
| id | integer | id | integer |
| dc_id | integer | dc_id | integer |
| id_src | text | id_src | text |
| polygon | geometry | polygon | geometry |
| vegetation type | text | parameter | text |
| start_date | date | timestamp | datetime |
| end_date | date | location | text |

For each of the 28 Sentinel-2 tiles, process 1 in the workflow (Figure 2) was to set a time range for which to perform the calculations, and to perform an SQL query to select all distinct tile geometries in table format from the Nexus database. Based on this time range and tile geometries, a nested for-loop was initiated (indicated by the yellow box in Figure 2), to be able to execute the iterations for all tiles in the study area and days in the selected time range. The first step in that loop, or process 2 in the overall workflow, was to download a number of Sentinel-2 images including their metadata from the database in the form of spectral bands (Green, Red, lower Red-Edge, upper Red-Edge, *NIR*), and a scene classification (SCL) band, a raster with categorical classes such as cloud shadow, cloud probability, water, cirrus, snow, vegetated pixels and unvegetated pixels [40]. All data were available as level-2A products containing Bottom-Of-Atmosphere (BOA) reflectance orthomosaics. Those level-2A products are generated by ESA by performing a pre-processing algorithm (Sen2Cor) on level-1C, or Top-Of-Atmosphere (TOA) products, based on a scene classification, and atmospheric, terrain and cirrus correction [23,41].

Process 3 in the workflow (Figure 2) was to resample some of the downloaded images to make sure all bands had a spatial resolution of 10 m. For that purpose, the SCL band was resampled by a nearest neighbor algorithm useful for discrete data, while the spectral bands were resampled by a bilinear interpolation, which is more suitable for continuous data [42]. Next, a cloud-masking algorithm was performed on the resampled spectral bands (process 4) to mask out pixels that were influenced by medium and high cloud cover probability and cloud shadow, based on a raster overlay with the SCL band. The masked spectral bands were used as input to calculate a number of vegetation indices applied more often for grassland monitoring: *NDVI*, *WDVI*, *NDRE* and $CI_{Red-Edge}$ (process 5). The *NDVI* (Normalized Difference Vegetation Index) was first introduced in 1979 [43], and has often been used for monitoring vegetation presence, for instance for evaluating ecological effects of environmental change on ecosystems [44]. It is calculated as a normalized difference between *NIR* and *Red* spectral wavelengths (Equation (1)).

$$NDVI = \frac{NIR - Red}{NIR + Red} \tag{1}$$

The *WDVI* (Weighted Difference Vegetation Index) was first developed as a model to estimate Leaf Area Index (LAI), and is determined by including the contribution of the soil reflectance ($r_{s,NIR}$, $r_{s,Red}$) to the vegetation reflectance (*NIR*, *Red*), assuming that the ratio between infrared and red reflectance of bare soil is constant (Equation (2)) [45].

$$WDVI = NIR - \frac{r_{s,\ NIR}}{r_{s,\ Red}} \times Red \tag{2}$$

Since soil reflectance is filtered from the signal, the differences between scarce and abundant vegetation are more obvious for the *WDVI* than for the *NDVI*, which is a valuable tool for assessing differences in vegetation properties [46]. In addition, two vegetation indices useful to quantify leaf chlorophyll content and biomass in grassland crops were implemented in the system as well [47,48]. The first one was the *NDRE* (Normalized Difference Red-Edge), whose equation is very similar to the *NDVI*, except that it uses the Sentinel-2 lower Red-Edge band ($RE_{band5}$) instead of the visible red band (Equation (3)).

$$NDRE = \frac{NIR - RE_{band5}}{NIR + RE_{band5}} \tag{3}$$

Secondly, the Red-Edge$_{chlorophyll}$ index ($CI_{Red-Edge}$) is based on a steep increase in reflectance between *Red* and *NIR* and calculates the ratio between the lower Red-Edge and upper Red-Edge bands (Equation (4)). Hence, it is a useful parameter to quantify leaf chlorophyll content in certain crops [49,50]. In its equation, $RE_{band7}$ and $RE_{band5}$ represent

Sentinel-2 bands with wavelengths in respectively the higher and lower region of the Red-Edge slope in the multispectral range.

$$CI_{Red\_Edge} = \frac{RE_{band7}}{RE_{band5}} - 1 \tag{4}$$

For each iteration in the workflow, all four most recently calculated vegetation index images were uploaded to the Nexus server and are visualized on StellaSpark's web-based user interface (indicated by process 8 in the flowchart in Figure 2).

In order to calculate zonal statistics for each grassland parcel based on these vegetation index images, first a SQL query was executed to select all parcels located within each Sentinel-2 tile (process 6 in Figure 2). This query consisted of a spatial overlay between the Sentinel-2 tile geometries and the grassland parcel geometries derived from the Dutch parcel database (BRP) that is linked to the Nexus platform. Based on this spatial overlay, zonal statistics in the form of mean, standard deviation and cloud cover percentage were calculated and stored in a local table (process 7). Cloud cover percentage was calculated by dividing the number of cloud-masked pixels by the total number of pixels covering one grassland parcel. Parcels containing a cloud cover proportion larger than a given threshold were discarded from the table, and were not used in the consecutive process of generating vegetation index time series. A lower cloud cover threshold led to fewer but more accurate data, while a higher percentage kept more data with higher inaccuracy probabilities. For the Sen2Grass algorithm, it was possible to manually adapt the maximum desired cloud cover percentage, in order to find a trade-off between data availability and data accuracy, and for the case study in this paper, a cloud cover threshold of 50% was chosen. At the end of each iteration, the zonal statistics table was pushed back to the Nexus database and appended to the existing table containing parcel statistics from previous iterations. The information in this table was used to generate time-series containing the mean, standard deviation and cloud cover percentage for each grassland parcel within each of the 28 Sentinel-2 tiles shown in Figure 1, and indicated by process 8 at the bottom of the flowchart in Figure 2.

### 2.3. Haus Riswick's Case Study: Field-Specific Analyses to Test the Sen2Grass Algorithm

As mentioned in Section 2.1, the Sentinel-2 tiles also cover a part of west Germany, including the city of Kleve in the state of North Rhine-Westphalia (NRW). Just to the east of that city, one of the two experimental and educational institutions of NRW's Land-wirtschaftskammer (Chamber of Agriculture), called Haus Riswick, is located. Some of Haus Riswick's main activities include the training of farmers, students and consultants in cattle breeding, developing novel technologies for improved dairy production and running trials on grassland parcels to improve crop yield while reducing environmental impact [51].

Similar to the earlier paper on this topic [33], three of Haus Riswick's experimental grassland parcels (Figure 3) were taken into account as a case study to test the Sen2Grass algorithm. Apart from remote sensing data from the Sentinel-2 constellation and parcel boundary data derived from the Nexus platform, also data on crop yield (in tons dry matter per hectare) for each of the three parcels for the years 2016 and 2018 were provided. Additionally, temperature and rainfall data from that region for the same years were available, which were extracted from a data log from a weather station near Kleve (top image of Figure 4). The weather data were used to give an explanation on possible causes for variations in grassland growth for the different growing cycles. In addition, the purpose of the crop yield data was to establish relationships between those data on the one hand, and vegetation indices on the other hand, in order to explore whether vegetation indices could be a good predictor for grassland biomass and/or chlorophyll content.

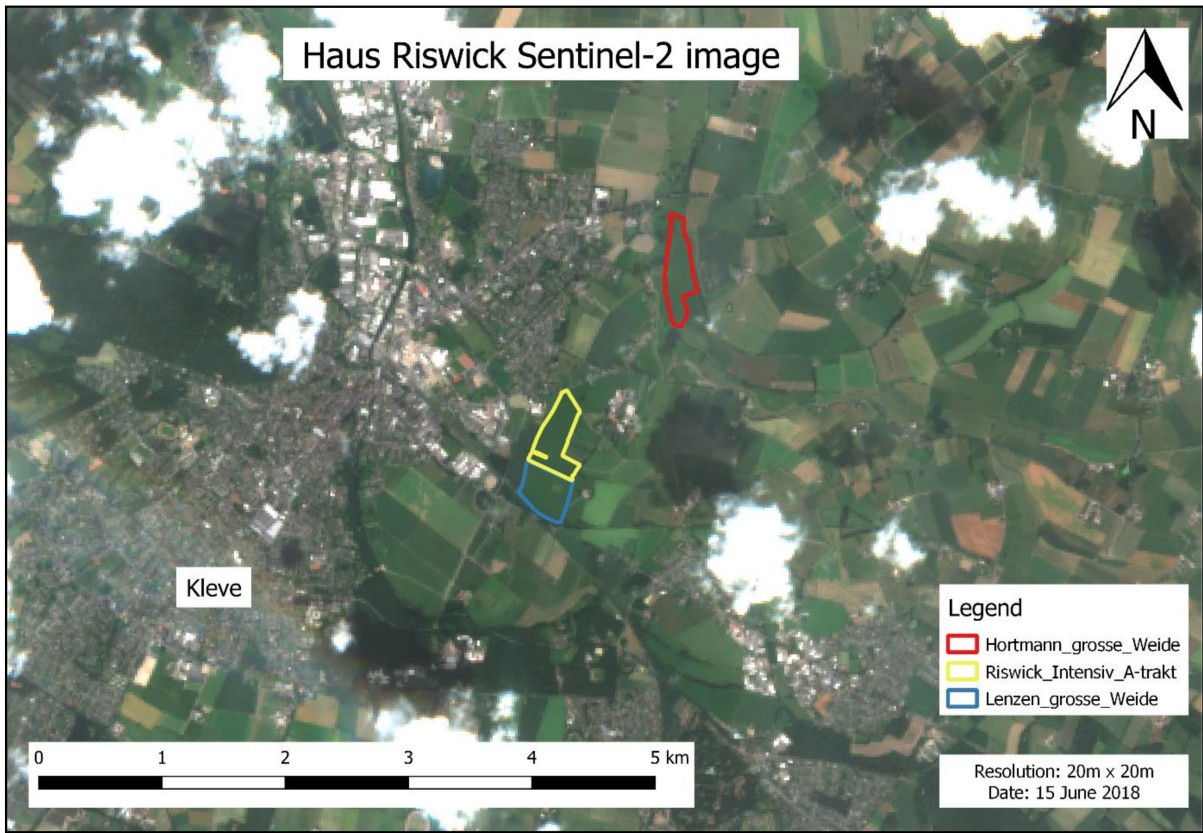

**Figure 3.** North Rhine-Westphalia (NRW)'s region where Haus Riswick is located (east of the city of Kleve). The orthophoto shows the three polygon objects of the evaluated experimental grassland parcels: Hortmann grosse Weide (most northern), Riswick Intensiv A-trakt (middle) and Lenzen grosse Weide (most southern).

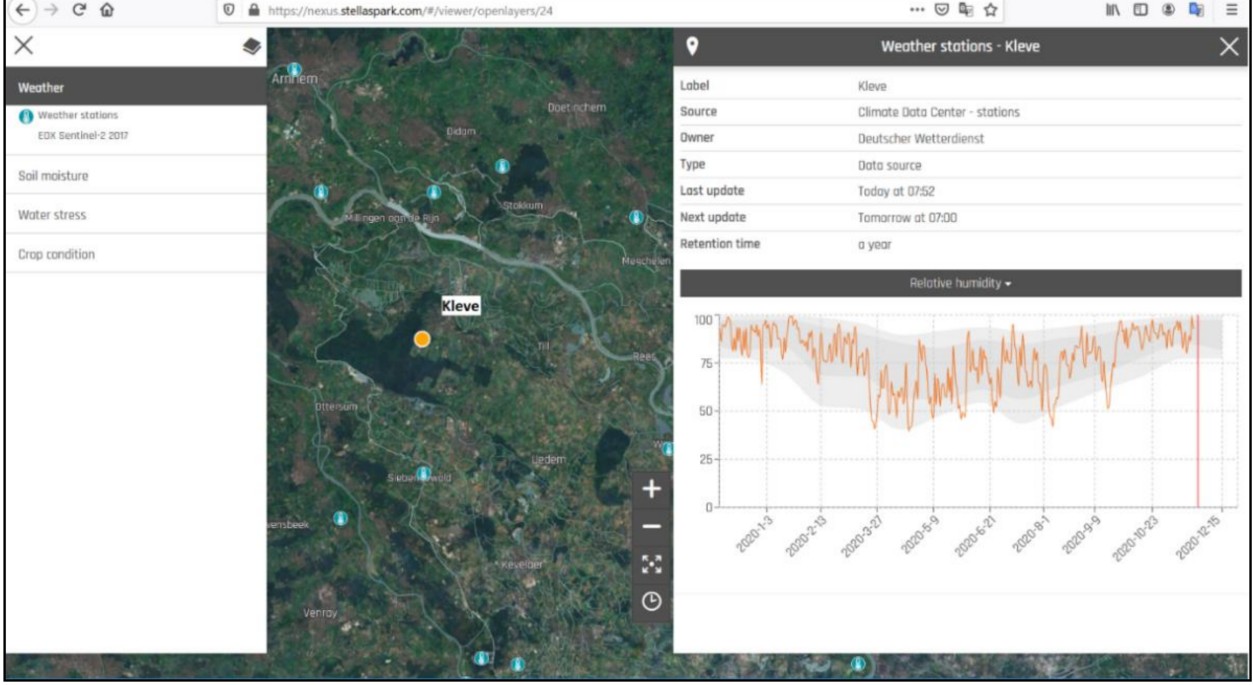

**Figure 4.** *Cont.*

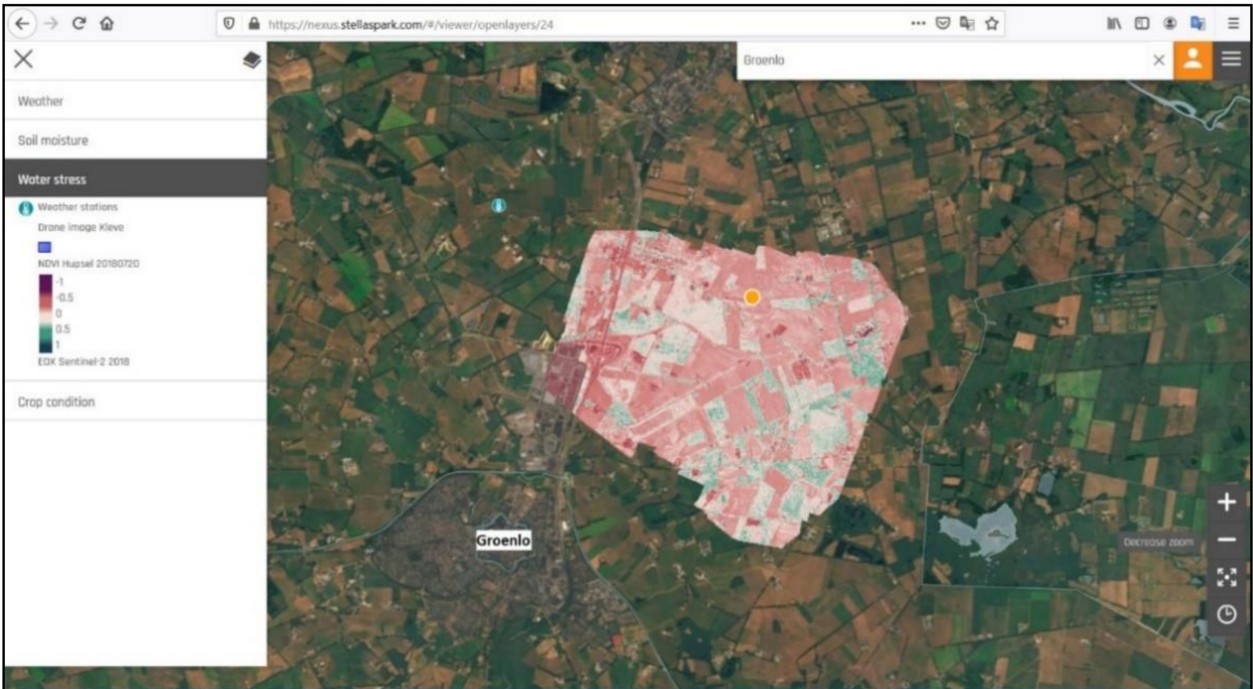

**Figure 4.** The Nexus web-based Graphical User Interface, showing time series depicting relative humidity based on data coming from a weather station close to Kleve, Germany (**top** image), and a *NDVI* image close to Groenlo in the east of the Netherlands (**bottom** image).

## 3. Results

### 3.1. Visualizations in StellaSpark's Nexus Platform

Graphical output for many kinds of environmental phenomena is being created in the back end of the Nexus platform and is visualized in an online web portal in an interactive way (Figure 4). Among other information, this includes information about drought monitoring, data coming from weather stations, and fertilizer application rates [35]. For instance, when clicking on the location of a weather station, in this example close to Kleve, Germany (top image of Figure 4), a pop-up menu like the one on the right of the screen will appear, showing in this case a time series of relative humidity for the area of Kleve between January and December 2020. Also, other visualizable weather parameters were available in that menu, such as daily precipitation and mean daily temperature, similar to the results for the case study of Haus Riswick presented in the next subsection. In addition, it was also possible to depict images of vegetation index on the Nexus web portal, such as the *NDVI* image for a region in the east of the Netherlands (bottom image of Figure 4), which was recorded in July 2018 with help of an Unmanned Aerial Vehicle (UAV) containing a camera with a very high spatial resolution.

Similar to the outputs in Figure 4, it is also possible to visualize the output from the Sen2Grass system on Nexus' web portal, based on the processed and calculated data that is being uploaded to the server. This information is twofold: on the one hand it consists of the most recent vegetation index images for each of the four calculated indices (Equations (1)–(4)) and for each of the 28 Sentinel-2 tiles shown in Figure 1. On the other hand, this includes a table with parcel statistics such as vegetation type, cloud cover percentage and mean and standard deviation for all four vegetation indices for all grassland parcels in one tile. The main field names of this table including data types are shown in Table 2, and its corresponding values are used as input to generate time series as portrayed in the top image of Figure 4 and in time series charts in the next subsection. Since the Sen2Grass processing chain is executed on a daily basis, the tables and the time series charts in Nexus are continuously being updated with the most recent calculations after finishing each iteration of the processing chain.

**Table 2.** Most important field names including data types of the parcels_new table containing calculations on vegetation index time series derived from Sentinel-2 data that are uploaded to the Nexus database.

| Parcels_New Table | | | | | |
|---|---|---|---|---|---|
| attribute | data type | attribute | data type | attribute | data type |
| id | integer | cloud_cover_perc | double | ndre_mean | double |
| id_src | text | *NDVI*_mean | double | ndre_std | double |
| tile_id | integer | *NDVI*_std | double | ci_red-edge_mean | double |
| time_stamp | datetime | wdvi_mean | double | ci_red-edge_std | double |
| vegetation type | text | wdvi_std | double | | |

### 3.2. Haus Riswick's Case Study: Field-Specific Results as Outcome of the Sen2Grass Algorithm

The Sen2Grass algorithm was tested on Haus Riswick's grassland parcels, and the mowing dates for the years 2016 and 2018, including corresponding crop yields (in ton dry matter per hectare) are listed in Table 3. This information shows that each growing season contained four cut-off dates and thereby contained four growing cycles. In most cases, a season's first growing cycle provided the highest amount of yield, while later growing cycles gave smaller amounts of yield. These data are important indicators to explain the vegetation index and time series patterns in the subsequent figures. For instance, Figure 5 shows Red-Edge$_{chlorophyll}$ index ($CI_{Red-Edge}$) images of Haus Riswick's grassland parcels for four selected dates of the growing seasons in 2016 and 2018. The $CI_{Red-Edge}$ values roughly ranged between 0 and 5, where red colors representing values closer to 0 indicate poor vegetation, while green colors specify values closer to 5, a sign of abundant vegetation. The images are shown each time in pairs, where the information in the first image (for instance Figure 5a) was recorded right before a cut-off date according to Table 3, while the information in the next image was recorded not long after that date of harvest (Figure 5b). This is highlighted by respectively high and low $CI_{Red-Edge}$ values that can be observed in those images. Similar patterns are observed in Figure 5c–l, which form the other pairs of images. For the field Lenzen grosse Weide, a noteworthy observation is a recurring different color pattern in the north-east of that parcel (Figure 5i–l), which can be explained by the presence of experimental plots on that location that had a different mowing regime than the rest of the parcel. Although the images in Figure 5 appeared to be completely cloud-free, images recorded on other dates could have been hindered by cloud shadow or cloud cover. As explained in Section 2.2, cloud shadows or cloud cover would lead to pixels being masked from the Sentinel-2 images, based on the scene classification (SCL) band, and therefore the cloud-cover threshold concept was introduced. This becomes more clear in Figure 6, where the Sentinel-2 images at the location of parcel Lenzen grosse Weide were hindered by cloud shadows (Figure 6b) or were fully covered with clouds (Figure 6c), respectively. Hence, the corresponding Sentinel-2 bands of those two dates (3 October and 12 December 2018) would be discarded for further calculations in the Sen2Grass algorithm. On the other hand, the Sentinel-2 data recorded on 15 June 2018 (Figure 6a) were completely cloud-free, and would therefore be included in the consecutive computations of the algorithm.

Figure 7 shows a collection of time series for Lenzen grosse Weide's parcel for the growing seasons in 2016 and 2018, that were constructed based on information included in Tables 2 and 3, and a third table containing meteorological information that was extracted from a weather station near Kleve (top image of Figure 4). Figure 7a,b illustrate the mean $CI_{Red-Edge}$ values and its corresponding standard deviations, and also the cut-off dates including crop yields according to Table 3 are presented in those charts. As expected, at most of the positions of these cut-off dates in the chart, large decreases in $CI_{Red-Edge}$ time series values are observed, which is an indication of the beginning of a new growing cycle. However, also sudden declines in $CI_{Red-Edge}$ values are visible at other positions in the chart, which partly could have been caused by influences of cloud cover in the data, since higher cloud cover percentages lead to larger calculation inaccuracies when determining a parcel's mean vegetation index. For the workflow described in this paper, a cloud cover threshold of 50% was chosen, indicated by the dashed horizontal lines in Figure 7e,f. As also mentioned before, data with cloud

cover percentages above these lines (the gray vertical bars) were discarded, while data with lower cloud cover percentages (the brown bars) were included for further computations in the Sen2Grass algorithm. At a selected cloud cover threshold of 50 percent, the proportion of discarded data was approximately 60 percent for both years; hence the remaining 40 percent of data were used to construct vegetation index time series as shown in Figure 7a,b. Lastly, a pattern with very low $CI_{Red\text{-}Edge}$ values is visible for the months July and August 2018 in Figure 7a,b, which was caused by a combination of a very hot and dry summer in the months of June, July and August in 2018, while 2016 was a wetter and cooler year. Similar patterns were observed in the graphical outputs of the other three vegetation indices, and the temperature and precipitation sequences in Figure 7c,d, showing the daily precipitation and mean daily temperature trends for the area of Kleve for the growing seasons of 2016 and 2018, confirm these tendencies.

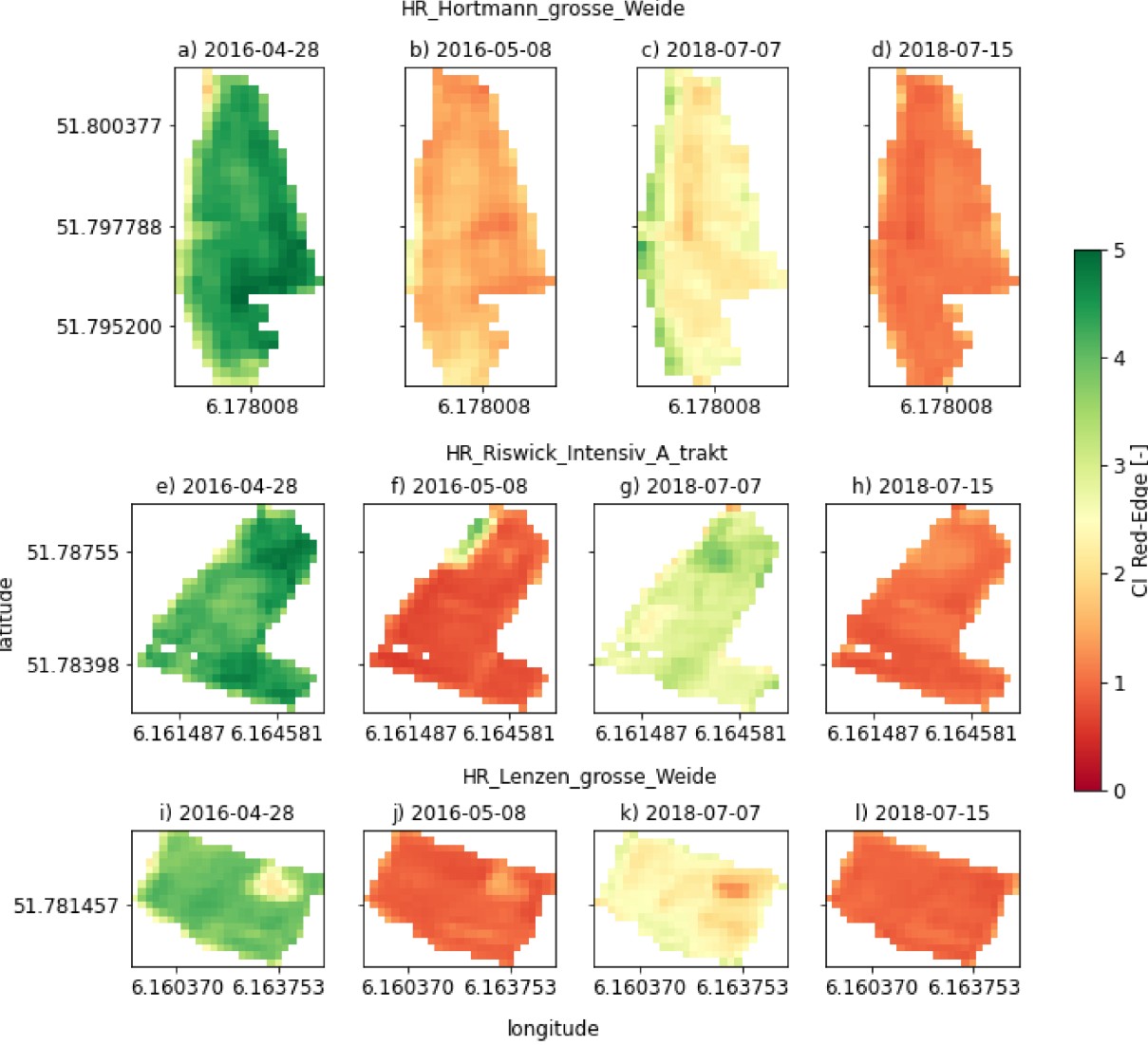

**Figure 5.** Vegetation index images ($CI_{Red\text{-}Edge}$) for the three parcels Hortmann grosse Weide (**a–d**), Riswick Intensiv A-trakt (**e–h**), and Lenzen grosse Weide (**i–l**), which are shown in pairs: just before and right after harvest for selected dates in the growing seasons of 2016 and 2018.

**Table 3.** Mowing dates and corresponding yield for the three grassland parcels of Haus Riswick for the years 2016 and 2018.

| Parcel | 2016 | Dry Matter Yield [t/ha] | 2018 | Dry Matter Yield [t/ha] |
|---|---|---|---|---|
| Hortmann grosse Weide | 5 May | 19.78 | 27 Apr | 23.31 |
| | 6 Jun | 34.14 | 26 May | 18.99 |
| | 20 Jul | 29.30 | 11 Jul | 9.23 |
| | 25 Aug | 26.93 | 9 Oct | 7.58 |
| Riswick Intensiv A-trakt | 4 May | 38.91 | 26 Apr | 29.67 |
| | 5 Jun | 22.76 | 26 May | 19.94 |
| | 19 Jul | 29.01 | 12 Jul | 18.47 |
| | 26 Aug | 25.42 | 9 Oct | 4.52 |
| Lenzen grosse Weide | 4 May | 31.16 | 26 Apr | 24.90 |
| | 5 Jun | 32.51 | 26 May | 24.03 |
| | 19 Jul | 25.25 | 12 Jul | 13.53 |
| | 26 Aug | 27.98 | 9 Oct | 8.62 |

The last goal of this paper was to establish relationships between vegetation indices on the one hand and crop yield on the other hand, in order to evaluate whether vegetation indices could be a good predictor for grassland biomass content. These relationships are depicted in the scatterplots of Figure 8. Anticipating that dry matter content would increase proportional to the increase of the $CI_{Red\text{-}Edge}$, positive trends were expected in the data. For most of the charts this was the case to some extent, since low to moderate $R^2$ values were observed (Figure 8b–d), whereas the trend for the parcel Hortmann grosse Weide seemed to be rather weak (Figure 8a). Similar patterns were observed in the associations between dry matter yield and the other three calculated vegetation indices that were described in Section 2.2.

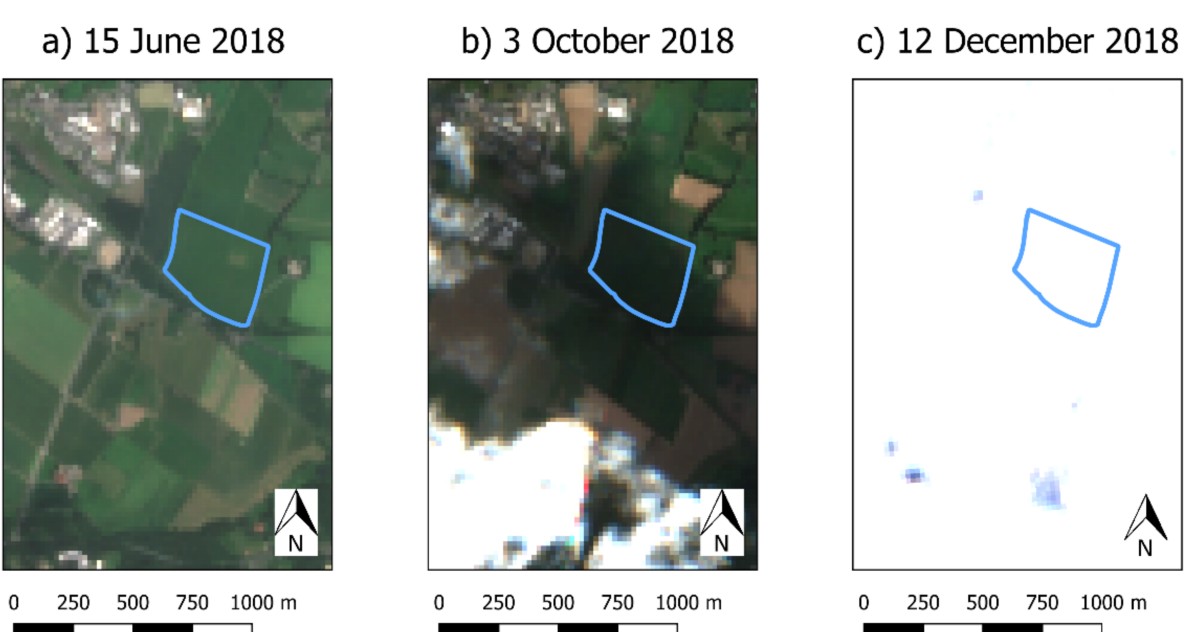

**Figure 6.** Cloud cover over Haus Riswick's Lenzen grosse Weide parcel: cloud-free (**a**); hindered by cloud shadows (**b**); and fully covered with clouds (**c**).

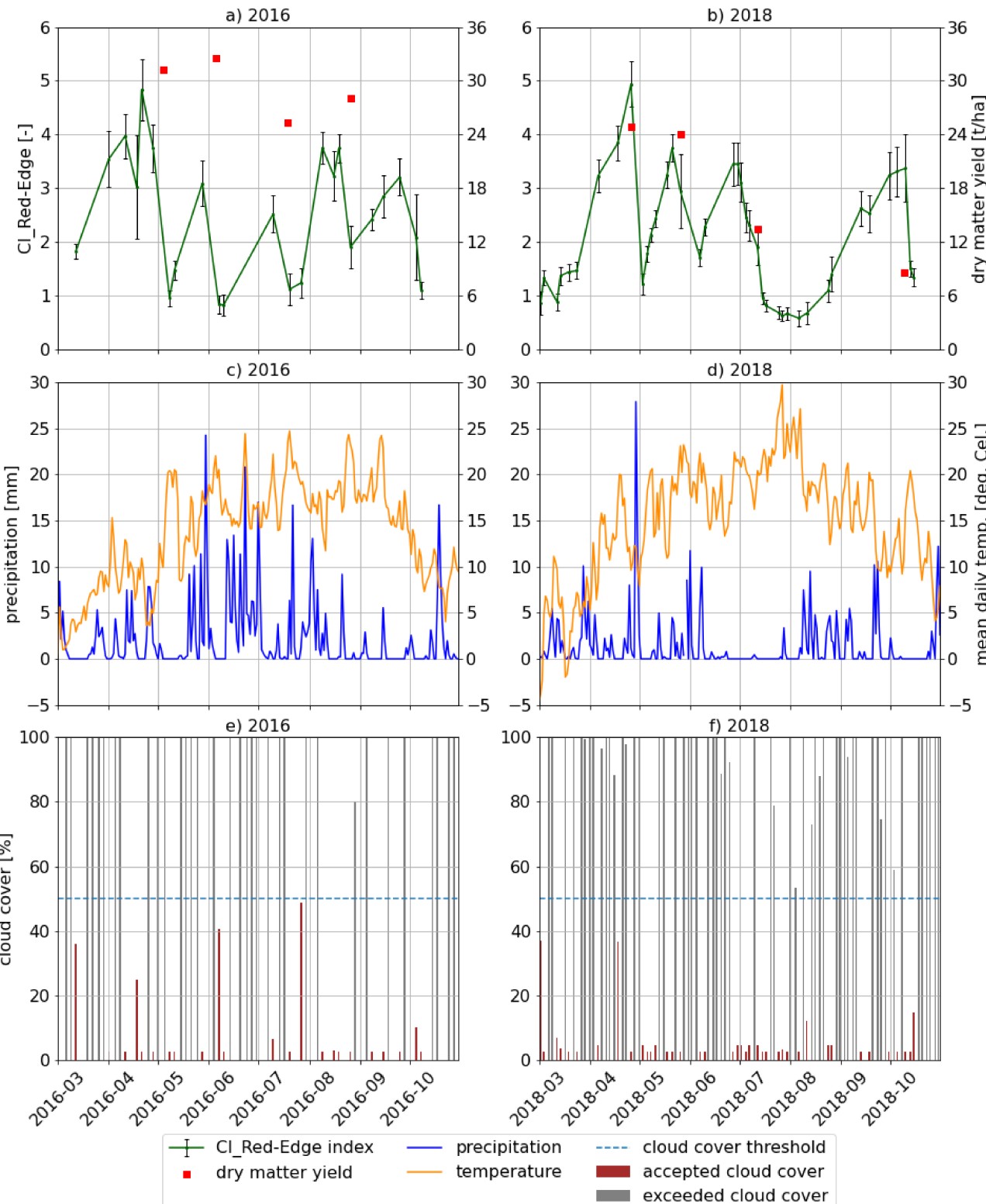

**Figure 7.** Time series over growing seasons 2016 and 2018: $CI_{Red\text{-}Edge}$ index and dry matter yield for parcel Lenzen grosse Weide (**a**,**b**), total precipitation (mm) and mean daily temperature (deg. C) for the area of Kleve (**c**,**d**), and cloud cover percentages for parcel Lenzen grosse Weide (**e**,**f**).

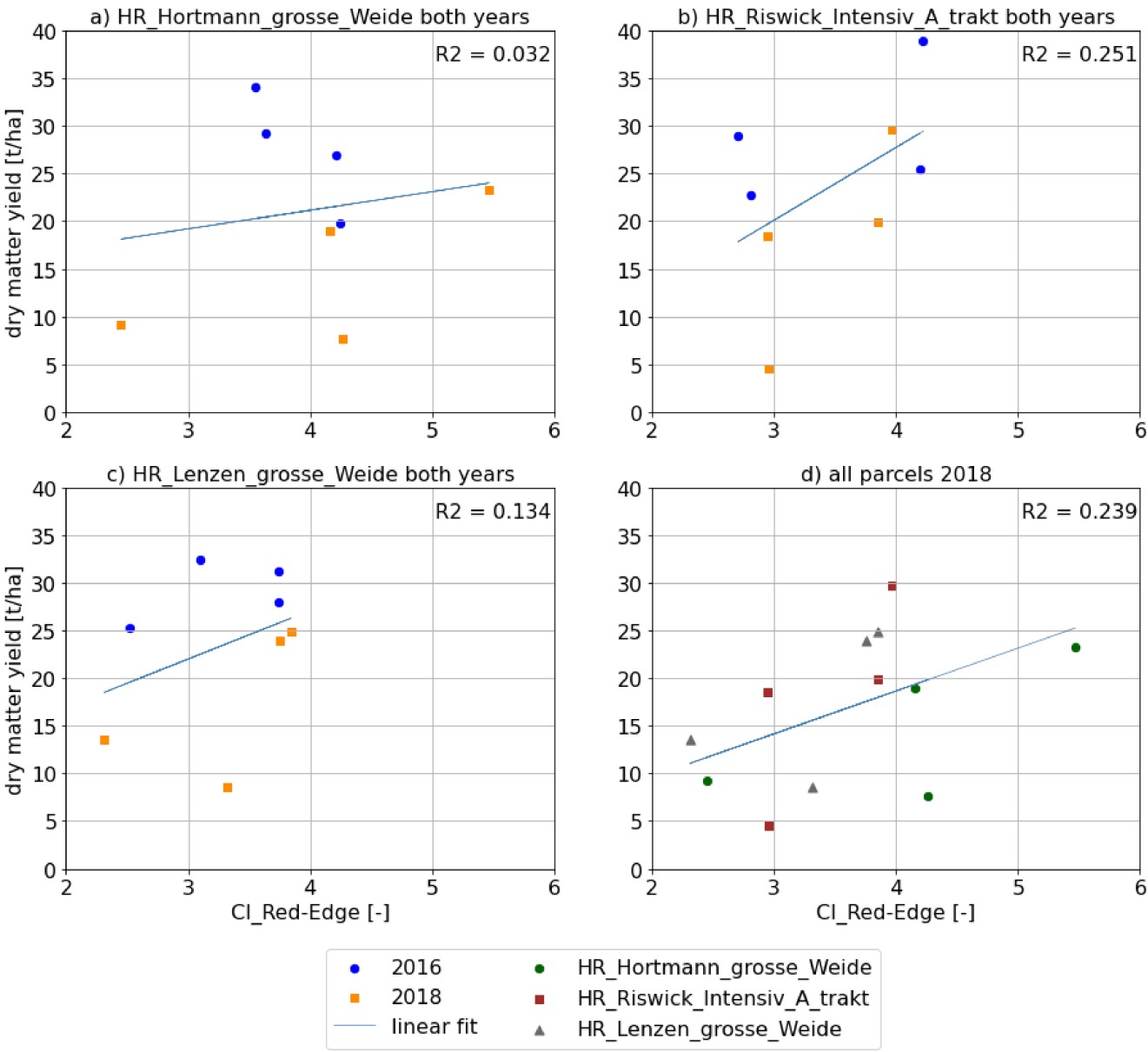

**Figure 8.** Scatterplots $CI_{Red-Edge}$ index vs dry matter yield for the parcels Hortmann grosse Weide, Intensiv A-trakt, Lenzen grosse Weide for the years 2016 and 2018 combined (**a–c**), and for all three parcels combined for the year 2018 (**d**).

## 4. Discussion

The main purpose of this research was to develop and test an automated and open-source grassland monitoring system (Sen2Grass) with a processing chain to harvest, process and analyze Sentinel-2 satellite data. These data were used to generate field-specific grassland information in the form of maps and time series of vegetation indices, in order to support management practices in grassland production for dairy farming. An earlier paper described the development of a preliminary stand-alone version of this system, making use of a limited dataset from Haus Riswick's experimental farm and the region around Kleve, Germany [33,51]. However, since a local computer's storage capacity and processing memory are limited, there was a need to upscale this system to an online cloud-computing environment to be able to generate field-specific grassland information on the national and regional level in a fast and automated way, which was presented in this paper.

For that purpose, the Spectors project partners decided to develop their own automated processing algorithm for grassland monitoring (Sen2Grass). The intention was to

do this independent from commercially-based cloud computing platforms such as AWS, Microsoft Azure and Google Earth Engine, in order to have more control on data processing and storage capacity, which sometimes could be a limitation in the usability of those platforms [27]. Therefore, the decision was made to collaborate with StellaSpark, a company that has been developing a web-based platform called Nexus: a cloud-computing environment with PostgreSQL/PostGIS functionalities to combine, integrate, maintain and visualize (geospatial) data streams from different sources into one platform [35]. For this research project, a study area containing the Netherlands, Belgium, and a part of France and Germany was chosen, covered by 28 Sentinel-2 tiles in total (Figure 1). However, Nexus is not limited in geographical scope, since StellaSpark provides solutions to create virtual environments with the size of a municipality, country, or continent, to be able to access and process (geospatial) data from any part of the world as inputs for an algorithm such as Sen2Grass.

Grassland parcel geometries were used to determine grassland statistics by performing an overlay between Sentinel-2 data and these geometries. However, a limitation in this process was the lack of parcel data availability for other countries than the Netherlands. While the Netherlands has been providing and annually updated and open-source parcel database on the national level (the Basisregistratie Gewaspercelen, or BRP) since several years, other countries often provide limited access to such data sources, or do not have them available at all. Therefore, solutions should be found to make those kinds of geospatial data sources more accessible in the future. Well-known examples of open-source geospatial data platforms are ESA's Copernicus Open Access Hub [24] and NASA's Open Data Portal [52]. Also, more and more governments on the local, national and international level have been providing so-called geodata-portals, which are web services to share, search, request, and obtain various kinds of geographic data and services [53]. Examples of national and international geodata portals are Europe's INSPIRE platform [54], US Geological Survey's Geo Data Portal [55] and the Dutch Publieke Dienstverlening op de Kaart (PDOK) [39].

The Copernicus Open Access Hub was the main data source to obtain the Sentinel-2 satellite data that was used to develop the Sen2Grass system. Despite the high spatiotemporal resolutions (10 m, 20 m and 60 m, and a revisit time of 2–3 days in mid-latitudes) [22], not all data were useful to calculate vegetation indices and to generate time series, since optical satellite sensors are often hindered by physical cloud cover, especially in humid areas in the world. This had an effect on image acquisition in such a way that certain pixels in an image were unusable for land-use monitoring. ESA has developed a tool (Sen2Cor) to atmospherically pre-process and correct Sentinel-2 data [23,41]. This tool uses the earlier described scene classification (SCL) band to distinguish physical cloud cover and cloud shadows from vegetated and unvegetated pixels (among other classes), to be able to systematically mask those cloud-influenced pixels from the acquired Sentinel-2 bands [23]. However, the more a surface area is covered by cloud cover pixels (Figure 6), the more uncertainty emerges in Sentinel-2's classification procedure that generates the SCL band, and the less reliable the remaining pixel values of an image become after performing a cloud-masking algorithm based on this classification band. Therefore, both in the earlier paper [33] and in this paper, the concept of a cloud cover threshold was introduced. In the first paper, a cloud cover threshold of 90% was selected, while in this paper a percentage of 50% was chosen. A higher cloud cover threshold led to more data availability, but also to more data unreliability and potentially more outliers, because of the higher classification inaccuracies of the land, water and cloud cover categories in the SCL band. On the other hand, a lower cloud cover threshold led to a scarcer dataset, but gave a larger probability for these data in terms of reliability, since the different categories in the SCL band were classified with higher accuracies. In conclusion, in order to find a suitable trade-off between data availability and data accuracy, the cloud cover threshold should be chosen in a customized way by any end-user of a grassland monitoring system such as Sen2Grass. Besides the selection of a cloud cover threshold, another way to deal with physical cloud cover influence would be to collect a sequence of images in time and systematically substitute cloud covered pixels by clean pixels from an

earlier or later image in a time series, for instance by using the Sen2Three tool, an algorithm that has been developed as part of ESA's Copernicus program as well [40]. However, this is not always desired, because this leads to a loss of data, since a sequence of images is often merged into one composite image. Lastly, also the decision could be made to plan missions to acquire high-resolution images with help of airborne-based platforms and substitute these images for Sentinel-2 data that were highly influenced by cloud cover. However, planning and flying such missions is often costly and time-consuming, especially because sunny, clear-sky conditions are required to acquire such data, while weather patterns are not always easy to predict in advance.

The Sen2Grass algorithm was tested on three experimental grassland parcels of Haus Riswick's experimental farm (Figure 3), and vegetation index time series turned out to be useful to detect different growing cycles within a growing season (Figure 5). Two additional goals for Haus Riswick's test case were to compare the generated vegetation index time series to meteorological patterns and to establish relationships between vegetation indices and dry grassland yield for the growing seasons of 2016 and 2018. First, the meteorological patterns provided one explanation for the presence of disparities in vegetation index patterns across time, since variations in grassland growth were influenced by fluctuations in temperature and precipitation, which was especially visible because of an extremely dry and hot summer in 2018 (Figure 7d). For the second goal, positive trends were expected in the relationships between vegetation indices and dry matter yield, and although some values suggested a positive association, results were not always consistent, because several $R^2$ values were very close to zero, and for some associations even negative trends were observed. A possible explanation for these poor results was the limited amount of data that were available on crop yield: only four cut-off moments per parcel per growing season; hence just 24 samples were obtained in total for all three fields for the growing seasons in 2016 and 2018 combined (Table 3). Therefore, no significant conclusions could be drawn based on these outputs on whether vegetation index time series would be a good predictor to estimate grassland biomass and/or chlorophyll content, as opposed to two studies that were successful in relating spectral reflectance to plant biomass and chlorophyll content [47,48]. One solution to improve these results would be to collect more grassland samples covering a larger land surface area and/or larger periods over time. A second method would be to develop and investigate different and more complex prediction models, which was carried out for instance for a study to predict plant diversity in grasslands based on time series from Sentinel-1 and 2 sensors and machine learning algorithms such as K-nearest neighbors and Random Forest models [14].

However, while the mowing dates across the growing seasons of 2016 and 2018 were provided for the case study of Haus Riswick (Table 3), this is often not the case for other grasslands. Therefore, other methods have been proposed to estimate cut-off dates, such as the automatic detection of mowing events across a growing season. For instance, two studies developed (machine learning) algorithms to reveal mowing frequencies from Sentinel-2 data based on the detection of significant deviations from *NDVI* time series [25,26]. Other methods to detect mowing events would be to include Synthetic-aperture radar (SAR) images recorded with the Sentinel-1 constellation instead of, or in addition to Sentinel-2 images, in order to develop more consistent and accurate time series based on data from these platforms [56–58]. Although the implementation of these kinds of methods still needs further development, this is expected to improve significantly in the years to come, because more and more spatiotemporal data with higher spatiotemporal resolutions, and from a wider variety of sources and platforms will become available to make these methods more robust and reliable in the future.

## 5. Conclusions

The main goal of this study was to further develop and test a primary grassland monitoring system called Sen2Grass, to determine field-specific grassland information on the national and regional level, based on data from Sentinel-2 satellite sensors, in

order to support more efficient and sustainable grassland management practices. This system has been implemented and tested with data from an experimental farm (Haus Riswick) in Nexus, an online storage and processing platform developed by StellaSpark for integrating, processing and visualizing large-scale data streams. The combination and implementation of different data sources led to a better understanding of a grassland ecosystem at the field scale. The results showed that the calculation of vegetation indices across time was useful for visualizing growing cycles in grassland production throughout a growing season, given that a maximum cloud cover percentage would not be exceeded. Moreover, meteorological data were useful to explain crop growth variations across these growing cycles over time. While mowing dates and crop yields were known for Haus Riswick, the inclusion of Sentinel-1 images could be a valuable addition to automatically detect mowing events for other grasslands as well. For the case study of Haus Riswick, the relationships between vegetation indices and dry grassland yield were poor to moderate. Because these relationships were based on a very small dataset, no valid conclusions could be drawn whether vegetation indices could be a potential predictor for grassland biomass and/or chlorophyll content. However, the inclusion of more and larger datasets could lead to develop more robust and reliable prediction models for that purpose in the future.

**Author Contributions:** T.H. and L.K. designed the research. L.K. supervised the research, conceived and managed the project. T.H. conducted the computational analysis, developed the script of the Sen2Grass algorithm, and wrote the final version of this manuscript. M.D.F. reviewed the initial computational analysis and provided the meteorological data for the area of Kleve. S.R. provided all other geographical data from Haus Riswick. E.V. developed the StellaSpark Nexus platform and together with D.v.D. supported the design of the Sen2Grass algorithm. G.v.d.E. and D.v.D. provided access to the Nexus platform through their company KnowH2O. All authors have read and agreed to the published version of the manuscript.

**Funding:** This work was supported by the SPECTORS project (143081), which was funded by the European cooperation program INTERREG Deutschland-Nederland.

**Informed Consent Statement:** Not applicable.

**Data Availability Statement:** The Python script to develop the Sen2Grass system that was described in this paper consists of two parts: one part was dedicated to developing a processing chain to be integrated in the Nexus platform, while the other part focused on the workflow and data analysis for the case study of Haus Riswick's experimental farm. Both parts of the script have been published on www.github.com/tomhardy084 (accessed on 1 March 2021), in the Spectors Project repository [34].

**Acknowledgments:** We thank ESA for offering the Copernicus Open Access Hub to be able to obtain data from the Sentinel-2 satellite constellation; Haus Riswick for offering us the data that we used for our case study; and StellaSpark for providing us the Nexus cloud computing platform to test and implement the Sen2Grass monitoring system.

**Conflicts of Interest:** The authors declare no conflict of interest. The funders had no role in the design of the study; in the collection, analyses, or interpretation of data; in the writing of the manuscript, or in the decision to publish the results.

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
