# Peer review of "Sen2Grass: A Cloud-Based Solution to Generate Field-Specific Grassland Information Derived from Sentinel-2 Imagery"

_agriengineering, doi:10.3390/agriengineering3010008_

Round 1

Reviewer 1 Report

 The study 'Sen2Grass: A cloud-based solution to generate field-specific 2 grassland information derived from Sentinel-2 imagery'  aimed to develop and test a primary grassland monitoring system called Sen2Grass, to determine field-specific grassland information on the national and regional level, based on data from Sentinel-2 satellite sensors, in order to support more efficient and sustainable grassland management practices. This system has  been implemented in Nexus, an online storage and processing platform for integrating,  processing and visualizing large-scale data streams.

As the conclusions of this study state, although relationships between vegetation indices and dry grassland yield were poor to moderate, vegetation indices  could be a potential predictor for grassland biomass content, provided that sufficiently large datasets are available to develop more robust and reliable prediction models.

Author Response

First of all, we want to thank the reviewer for the suggestions to improve the manuscript. Below we elaborate in more detail how we took the suggestions into account.

  1. The study 'Sen2Grass: A cloud-based solution to generate field-specific 2 grassland information derived from Sentinel-2 imagery' aimed to develop and test a primary grassland monitoring system called Sen2Grass, to determine field-specific grassland information on the national and regional level, based on data from Sentinel-2 satellite sensors, in order to support more efficient and sustainable grassland management practices. This system has been implemented in Nexus, an online storage and processing platform for integrating, processing and visualizing large-scale data streams.

Reply: This is a general summary of our paper, so no edits have been made based on this comment.

  1. As the conclusions of this study state, although relationships between vegetation indices and dry grassland yield were poor to moderate, vegetation indices could be a potential predictor for grassland biomass content, provided that sufficiently large datasets are available to develop more robust and reliable prediction models.

Reply: Thanks for this suggestion, we have updated the conclusion section on this point. In lines 614-620, a clarification was made that the relationships between vegetation indices and dry grassland yield for the case study of Haus Riswick were poor to moderate. Because these relationships were based on a very small dataset, no valid conclusions could be drawn whether vegetation indices could be a potential predictor for grassland biomass and/or chlorophyll content. However, the inclusion of more and larger datasets could lead to develop more robust and reliable prediction models for that purpose in the future.

Reviewer 2 Report

I like the idea, however I miss sth in that paper - Authors promise testing of the developed software, but nothing is presented - this step should be beneficial for the paper; some comparisons of the Author's development with so-far existing softwares should be done

the case study seems to be glued just to extend the paper, no connection with the large scale investigation

you introduced cloud cover threshold concept - an analyses should be done and presented for various initial values - the issue is discussed in "discussion" chapter, but it would be better to support it with some raw data

L13 rather "despite"

figure 1 & section 2.3:  I would appreciate labeling of the process steps both on the figure and in the text e.g.:  A next step in the workflow (#2, Figure 1). IMO it would increase the clarity and make it easier to follow

figure 3:  not "the map shows" - rather orthophoto/image/ortomosaic

section 3.1: - well, I don't think it is a result, for me that part should still be in "material and methods"

section 3.2 - don't describe what the figures literally show, focus on the meaning of what is shown e.g. figure 7 shows scatterplots - we see it - the important thing is that the relationship is rather weak - skip lines 356-365

"Discussion" - too much of the results repetition

L464 R2 had negative values???

Author Response

First of all, we want to thank the reviewer for the suggestions to improve the manuscript. Below we elaborate in more detail how we took the suggestions into account.

  1. I like the idea; however, I miss something in that paper - Authors promise testing of the developed software, but nothing is presented - this step should be beneficial for the paper; some comparisons of the Author's development with so-far existing software should be done.

Reply: In the discussion it has been made more explicit that we had the intention to develop the Sen2Grass algorithm independent from commercially based cloud computing platforms such as AWS, Microsoft Azure and Google Earth Engine, in order to have more control on data processing and storage capacity, which sometimes is a limitation for the usability of those commercially based platforms. Also, in connection with remark #2 below, we want to state that Haus Riswick’s data were used as a case study to test the Sen2Grass algorithm.

  1. The case study seems to be glued just to extend the paper, no connection with the large-scale investigation.

Reply: The case study of Haus Riswick illustrates what information can be derived from the Sen2Grass system at the field level, and therefore shows how it can be used to support grassland management practices. This has been made more explicit in the methodology, results, and discussion sections of the paper, where Haus Riswick’s grassland data have been applied as a case study to test the Sen2Grass algorithm. This is also made more explicit by changing the headings of section 2.3 and 3.2 accordingly.

  1. You introduced cloud cover threshold concept - an analysis should be done and presented for various initial values - the issue is discussed in "discussion" chapter, but it would be better to support it with some raw data.

Reply: In the methods, results and discussion sections, the why and how of the cloud cover percentages and threshold are described more elaborately: the more a surface area is covered by cloud cover pixels, the more uncertainty emerges in the Sentinel-2’s classification procedure that generates the SCL band, and the less reliable the remaining pixel values of an image become after performing a cloud-masking algorithm based on this classification band. Therefore, the concept of a cloud cover threshold was introduced, so that data with cloud cover percentages above this threshold could be discarded. Some extensive edits have been made on this issue in sections 2.2 (lines 287-292), and section 3.2 (lines 388-406 and 422-430). Moreover, this has been made more intuitive with an added Figure 6 that shows three different levels of cloud cover: cloud-free (a), cloud shadow (b), and full cloud coverage (c).

  1. L13 rather "despite".

Reply: “In spite of” in the abstract in line 13 is changed to “despite”.

  1. Figure 1 & section 2.3: I would appreciate labeling of the process steps both on the figure and in the text e.g.:  A next step in the workflow (#2, Figure 1). IMO it would increase the clarity and make it easier to follow.

Reply: The operations/processes in the flowchart in Figure 2 are now numbered, and references to these numbers are made in the main text.

  1. Figure 3: not "the map shows" - rather orthophoto/image/orthomosaic.

Reply: “The map shows…” is changed to “The orthophoto shows…”.

  1. Section 3.1: well, I don't think it is a result, for me that part should still be in "material and methods".

Reply: We have thought about this suggestion. The StellaSpark Nexus-based grassland monitoring workflow is one of the outcomes of the development cycle as presented in Figure 1. As such, we wanted to present some visualizations in the Graphical User Interface as a result, as it shows what the currently available information in the system looks like, and to give an indication on how the Sen2Grass information is going to look like as soon as this system is fully operational in the Nexus platform.

  1. Section 3.2: don't describe what the figures literally show, focus on the meaning of what is shown, e.g. figure 7 shows scatterplots - we see it - the important thing is that the relationship is rather weak - skip lines 356-365.

Reply: In the results section, we presented figures that show the development of grassland parcels over time, for two growing seasons (2016 and 2018) in this case. We intended to describe these results as complete as possible, for the reader to get a clear overview on what the outputs are telling them, while in the discussion section we elaborate more on the interpretation of these results.

  1. "Discussion" - too much of the results repetition.

Reply: The discussion has been checked again, some repetitive information has been deleted, and some sentences are rephrased. The main edits in the discussion were to explain why the Spectors project stakeholders decided to develop their ow system independent from commercial platforms such as AWS and GEE (lines 486-492), and to explain that Haus Riswick’s grassland parcels were used as a case study to test the Sen2Grass algorithm (lines 554-557).

  1. L464: R2 had negative values?

Reply: Not the R2 values itself were negative, but associations with negative trends were observed. This misconception has been changed in the discussion section in lines 568-569.

Reviewer 3 Report

As a general comment, the paper is well written and structured, with a clear purpose. In my opinion the paper can be published after minor changes.

There are few minor corrections in the following comments:

At the end of the introduction a paragraph is needed with the structure of the rest of the paper.

At lines 93-94 there is a reference for “the framework of the SPECTORS project” without any other information or a citation. A reference is presented later on line 122 only for the Gitlab repository of the project.

Section 2.1 has the “StellaSpark Nexus” title which is the framework used for the implementation. The section actually describes Sen2Grass, the open-source and automated monitoring system, which should have its name as title.

The title of Table 1 states that the attributes are used “as information for the SQL statements”. SQL statements take field names.

Tables and Figures should be numbered according to their order of reference. Table 3 is referenced at line 254 before Table 2 at line 289. The same for Figure 6 which is referenced at line 278 before Figure 5 at line 310.

Some attention should be given to the language of the text. E.g., The statement in lines 243-345 needs restructure. The two statements contained in lines 271 - 276 start with “for instance” and should be rephrased. Also, in line 292 “both this table” should be corrected. 

Author Response

First of all, we want to thank the reviewer for the suggestions to improve the manuscript. Below we elaborate in more detail how we took the suggestions into account.

  1. At the end of the introduction a paragraph is needed with the structure of the rest of the paper.

Reply: A paragraph to describe the paper’s structure is added at the end of the introduction (lines 118-123).

  1. At lines 93-94 there is a reference for “the framework of the SPECTORS project” without any other information or a citation. A reference is presented later on line 122 only for the Gitlab repository of the project.

Reply: In lines 93-96 it is clarified that the paper was written within the framework of the SPECTORS project, and a reference to the website of this project is added. The reference to the GitHub repository is independent from the website of the SPECTORS project. The link to the GitHub repository in line 131 is removed, but we left the reference, since it is part of the supplementary material.

  1. Section 2.1 has the “StellaSpark Nexus” title which is the framework used for the implementation. The section actually describes Sen2Grass, the open-source and automated monitoring system, which should have its name as title.

Reply: In order to have a more logical order of the methodology, section 2.1 (StellaSpark Nexus) is merged into section 2.2 (renamed to “Development of the Sen2Grass processing chain” in line 186), and the first subsection of the methodology is now the description of the Geographical focus of the algorithm.

  1. The title of Table 1 states that the attributes are used “as information for the SQL statements”. SQL statements take field names.

Reply: The word “attributes” is changed to “field names” in the table headings in lines 223-225 and 361-362, as well as in the main text in line 222 and line 353.

  1. Tables and Figures should be numbered according to their order of reference. Table 3 is referenced at line 254 before Table 2 at line 289. The same for Figure 6 which is referenced at line 278 before Figure 5 at line 310.

Reply: The reference to Table 3 in line 315 is removed, and the references to Figure 6 in lines 340 and 355 are rephrased to “The results for the case study of Haus Riswick are presented in the next subsection” and “…the time series charts in the next subsection…”, respectively.

  1. Some attention should be given to the language of the text. E.g., The statement in lines 243-245 needs restructure. The two statements contained in lines 271 - 276 start with “for instance” and should be rephrased. Also, in line 292 “both this table” should be corrected.

Reply:  Thanks for this suggestion, we put some extra attention to sentence structure and language. Based on this, the following edits have been made:

  • “Located east of that city…” in lines 303-304 is changed into “Just to the east of that city…”.
  • The sentence above Figure 3 (lines 317-323) is split in two, to use more concise language.
  • “This output includes for instance information…” in lines 333-334 is changed to “Among other information, this information includes information about…”.
  • The sentences in lines 356-360 are rephrased to use more clear language: “Since the Sen2Grass processing chain is executed on a daily basis, the tables and the time series charts in Nexus are continuously being updated with the most recent calculations after each iteration of the processing chain.”

Round 2

Reviewer 2 Report

Authors introduced corrections that really improved the paper and made it clear and sound - thanks

L182 "in table 1" missing here I guess